# Transcriptome Analysis and Screening of Genes Associated with Flower Size in Tomato (*Solanum lycopersicum*)

**DOI:** 10.3390/ijms232415624

**Published:** 2022-12-09

**Authors:** Yiyao Zhang, Aining Zhang, Wenhui Yang, Xinyi Jia, Qingjun Fu, Tingting Zhao, Jingbin Jiang, Jingfu Li, Huanhuan Yang, Xiangyang Xu

**Affiliations:** Laboratory of Genetic Breeding in Tomato, Key Laboratory of Biology and Genetic Improvement of Horticultural Crops (Northeast Region), Ministry of Agriculture and Rural Affairs, College of Horticulture and Landscape Architecture, Northeast Agricultural University, Harbin 150030, China

**Keywords:** tomato, flower development, RNA-seq, *Solanum lycopersicum*, bZIP family

## Abstract

Flower development is not only an important way for tomato reproduction but also an important guarantee for tomato fruit production. Although more and more attention has been paid to the study of flower development, there are few studies on the molecular mechanism and gene expression level of tomato flower development. In this study, RNA-seq analysis was performed on two stages of tomato flower development using the Illumina sequencing platform. A total of 8536 DEGs were obtained by sequencing, including 3873 upregulated DEGs and 4663 down-regulated DEGs. These differentially expressed genes are related to plant hormone signaling, starch and sucrose metabolism. The pathways such as pentose, glucuronate interconversion, and Phenylpropanoid biosynthesis are closely related and mainly involved in plant cellular and metabolic processes. According to the enrichment analysis results of DEGs, active energy metabolism can be inferred during flower development, indicating that flower development requires a large amount of energy and material supply. In addition, some plant hormones, such as GA, may also have effects on flower development. Combined with previous studies, the expression levels of *Solyc02g087860* and three of *bZIPs* were significantly increased in the full flowering stage compared with the flower bud stage, indicating that these genes may be closely related to flower development. These genes were previously reported in Arabidopsis but not in tomatoes. Our next work will conduct a detailed functional analysis of the identified bZIP family genes to characterize their association with tomato flower size. This study will provide new genetic resources for flower formation and provide a basis for tomato yield breeding.

## 1. Introduction

As the reproductive organs of angiosperms, flowers have various structural changes. For a long time, the development of floral organs has attracted much attention from researchers. There are the ABC model, ABCDE model, four-factor model, etc. Sepals, petals, stamens and carpel constitute the four-wheel flower organs of tomato [1]. In 1991, Coen et al. proposed an ABC model of floral development that could account for homeotic mutations [2]. With the further study of genes regulating flower development, more flower homeotypic genes have been cloned. In 1995, scientists discovered two genes (*FBP7* and *FBP11*) regulating ovule development, which was found to belong to a new class of MADS-box genes for the normal development of morning glories ovule, so these genes were classified as class D genes [3,4]. Later studies found that *AGL11* (*STK*) gene is homologous to *the FBP11* gene and belongs to the class D gene with *SHP1* and *SHP2* [5]. When studying the ABC model of flower development, it was found that flower organs could be transformed from leaves, but the ABC model could not complete the regulation. When Mizukami et al. studied that all four-wheel flower organs could be transformed from leaves, they found that when *AGL2* (*SEP1*), *AGL4* (*SEP2*) and *AGL9* (*SEP3*) genes of Arabidopsis were mutated simultaneously, all flower organs only formed sepals [6]. Therefore, these genes are classified as class E genes, and the ABCDE model of flower development is proposed.

*MC* genes are class A genes found in tomatoes [7]. It was found that the inflorescence of the Mc-vin mutant recovered vegetative growth after the formation of 2–3 flowers, indicating that the mutant gene is crucial for maintaining the characteristics of the inflorescence meristem. Moreover, the *MC* gene is also part of the genetic network regulating the development of tomato inflorescence meristem, and the gene also regulates the development of sepals [8]. The *SlGLO1* gene in tomatoes is a class B gene. Geuton et al. found that the stamens and petals of tomato were transformed into carpel and sepal, respectively, after silencing the *SlGLO1* gene, and the expression of *SlGLO2* and *TM6* genes in the flower organs of tomato with *SlGLO1* gene silencing was reduced in the second and third round of flower organs. The expression of class D, class E and class F genes also decreased in the third round of flower organs, and the results showed that *the SlGLO1* gene could positively regulate the other three class B genes, and the class B genes could be cross-regulated [9]. Guo et al. found that the *SlGLO1* gene was highly expressed in petals and stamens, and RNAi inhibition of *SlGLO1* gene expression resulted in phenotypic abnormalities of flower organs, including green petals, shorter petals, abnormal pistils and male sterility. The research results showed that *the SlGLO1* gene played an important role in regulating tomato flower organs and pollen development [10]. Yang et al. identified a novel stameneless tomato mutant in which the development of stamens and carpel was disturbed. The gene responsible for the mutation is *SlGT11*, which is expressed in the early flower stage and, more specifically, in the primordium position corresponding to stamens and carpel in the later flower development. *SlGT11* gene silencing by RNAi was used to verify the phenotype of the mutant, and it was found that *SlGT11* gene silencing resulted in the loss of floral organ characteristics and made the flower transition to vegetative development during flower development [11].

Besides their important roles in plant stress resistance, the bZIP TFs family has also been found to play important roles in plant growth and development. Amanda et al. found that *the AtbZIP9* gene is expressed in the phloem of all organs when studying the functional properties of the C group of bZIP transcription regulators in Arabidopsis. Further studies using transgenic technology showed that *the AtbZIP9* gene played a role in the phloem development of *Arabidopsis* [12]. Lee et al. found that *Arabidopsis* plant growth was inhibited, plant development was delayed, and the number of petals was reduced by over-expression of *the CabZIP1* gene, suggesting that this gene plays a negative regulatory role in plant growth and development [13]. Zou et al. showed that inhibition of *OsABI5* gene expression led to a reduction in rice yield, indicating that *the OsABI5* gene plays a regulatory role in rice reproduction [14]. Chuang et al. found that the mutation of the *PAN* gene in Arabidopsis transformed the four-petaled flower into the five-petaled flower. The *PAN* gene is a member of the bZIP family of Arabidopsis, and this encoding protein exists in the apical meristem, floral meristem, whorl organ primordium and ovule primordium during the development of the wild-type flower. The results of transgenic technology showed that *the PAN* gene affected the characteristics of flower meristem, the size of flower meristem and the expression of genes related to the number of flower organs [15]. Guan et al. isolated the bZIP family gene *CAREB1* from a cDNA library of somatic embryos of carrot (*Daucus carota* L.) cultured in high sucrose medium. *CAREB1* gene is an important trans-acting factor during the development of carrot somatic embryos. Moreover, it plays a regulatory role in the development of carrot embryos [16]. Liu et al. showed that the bZIP transcription factor *HY5* mediates *Cry1A*-induced anthocyanin synthesis in tomatoes [17]. Sagor et al. showed that tomato *SlbZIP1* and *SlbZIP2* could improve the sweetness of tomato fruit [18].

Transcriptome research can study gene function and structure at the overall level and reveal specific biological processes. It has been widely used in plant gene candidate discovery, functional identification, and genetic improvement. In this study, the wild-type tomato (*Solanum lycopersicum cv. Alisa Craig*, hereafter referred to as AC) was selected as the research material, and the sampling time was at the bud stage (about 1 cm) and the full flower stage (the petals were fully expanded). The differentially expressed genes (DEGs) were analyzed by RNA sequencing (Rna-Seq) platform. Candidate genes related to flower size were also mined and screened. Our main goal is to obtain new genes that regulate flower development and provide a new basis for tomato yield breeding.

## 2. Results

### 2.1. Quality Analysis of RNA-Seq Data

In order to obtain the transcriptome information of wild-type tomatoes at different flower development stages, we used Rna-Seq to sequence the flower samples of wild-type tomatoes at two stages of bud stage (CK1) and full flower stage (CK2), and each sample was performed three biological replicates (Figure 1). In this experiment, a total of 6 samples were measured using the DNBSEQ platform, and Table 1 shows the Rna-Seq data quality of the 6 samples. After data filtering, a total of 38.58 Gb of filtered base number was obtained. The average number of filtered bases obtained from CK1 samples was 6.45 Gb, and the average number of filtered bases obtained from CK2 samples was 6.41 Gb. The percentage of bases with an error rate less than or equal to 0.1% (Q30) of these 6 samples was more than 90.42%, which indicated that the Rna-Seq data quality of these 6 samples was good.

### 2.2. Analysis of Differentially Expressed Genes

To investigate the related genes in different flower development stages of wild-type tomatoes, we performed a standardized analysis of DEG Seq on the two sets of transcriptome data. The criteria for determining DEG were Q-value ≤ 0.001 and Log2 fold change ≥ 2. The differential expression results obtained by screening between groups were displayed by a volcano diagram, as shown in Figure 2, where red represents upregulated differentially expressed genes, and blue represents down-regulated differentially expressed genes. A total of 8536 differentially expressed genes were identified, including 3873 upregulated differentially expressed genes and 4663 down-regulated differentially expressed genes. Genes with the ability to encode transcription factors (TFs) were predicted. At the same time, the transcription factor families to which the genes belong were classified and statistically analyzed, and the results are shown in the classification map of the transcription factor families to which the genes belong. There were 31 genes belonging to the bZIP transcription factor family (Appendix A).

### 2.3. GO Analysis and KEGG Analysis of Differentially Expressed Genes

To investigate the function of differentially expressed genes during tomato flower development, we performed GO analysis (Figure 3 and Appendix A) and KEGG analysis (Figure 4 and Appendix A) of DEGs using the Phyper function of R software. In the Biological process category of the GO classification, the differentially expressed genes in the cellular process, metabolic process, biological regulation and response to stimulus were significantly enriched. In cellular component classification, differentially expressed genes in the cell, membrane, membrane part and organelle were significantly enriched. Regarding the molecular function category, the differentially expressed genes in the binding and catalytic activity categories were significantly enriched. Appendix A shows that bZIP family GO classification was mainly “molecular function” and was significantly enriched in DNA-binding transcription factor activity and transcription regulator activity. The biological pathways of tomato flower development were further studied by KEGG enrichment analysis. Figure 4 shows that KEGG pathway enrichment mainly includes plant hormone signal transduction, starch and sucrose metabolism (Metabolism), pentose, glucuronic acid conversion (Pentose and glucuronate interconversions), styrene acrylic biosynthesis (Phenylpropanoid biosynthesis), etc. The KEGG pathway enrichment of the bZIP family mainly includes Plant hormone signal transduction and MAPK signaling pathway-plant (Appendix A).

### 2.4. Metabolic and Regulatory Pathway Analysis of DEGs

MAPMAN software was used to analyze the transcriptome data, and the results of enrichment analysis were visualized to analyze the pathways where the differentially expressed genes were located. As shown in Figure 5 and Figure 6, most of the upregulated genes were related to photoresponse, photorespiration, TCA, secondary metabolism and nitrogen and sulfur metabolism. In addition, some genes in photosynthesis, photorespiration and other pathways were also affected, and their expression levels were down-regulated. IAA, ABA, GA, BA and other plant hormones play an important role in the development of plant flower organs, and most of the upregulated genes are related to GA.

### 2.5. qRT-PCR Validation of RNA-Seq Data

Eight differentially expressed genes were selected for real-time fluorescence quantitative verification. These eight DEGs were mainly selected from genes related to flower development. As shown in Figure 7, the data of qRT-PCR and the results of Rna-Seq conform to a similar trend, confirming the accuracy of the results of RNA-Seq. Among these eight DEGs, the *Solyc04g015530* (*PS-2*) gene was significantly upregulated during flower development, and the expression level was increased about 10-fold (Figure 7b). Among them, the class B genes *Solyc04g081000* (*TAP3*) and *Solyc08g067230* (*FBP1*) in the flower development model were upregulated to varying degrees during flower development (Figure 7a,e). The class C gene *Solyc02g071730* (*TAG1*) in the flower development model showed a slight decrease in expression during flower development (Figure 7f).

### 2.6. Screening of Candidate Genes

In Arabidopsis, bZIP family genes were found to be involved in the regulation of flower development. The bZIP family genes in transcriptome data were analyzed by expression clustering using BGI interactive reporter system (Figure 8), and 8 bZIP family genes with higher expression levels were selected for mapping through screening in this study (Figure 9). The results show that the coated compared with bud period, *Solyc02g062950* (*GBF12*), *Solyc04g077555*, *Solyc02g085610* (*GBF4*), *Solyc11g064953* these four genes expression was significantly decreased, increased *Solyc11g044560* (*ABF4*) expression was not significant. The expression levels of *Solyc07g053450* (*bZIP61*), *Solyc03g046440* (*bZIP43*) and *Solyc07g062710* (*bZIP61-like*) were significantly increased, which might be related to the development of flower organ in tomato. Among them, the expression level of *Solyc07g062710* (*bZIP61-like*) changed most obviously, and the flowering stage was about twice that of the flower bud stage.

## 3. Discussion

At present, transcriptome sequencing technology has been widely used in the study of plant development and other aspects. With the study of plant flower development and flowering time regulation, many genes related to flower development have been identified, which provides a theoretical basis for studying the molecular mechanism of plant flower development. Zhang et al. conducted RNA-Seq analysis when studying the *SlMYB33* gene mediating tomato flowering and pollen development. Through KEGG pathway analysis, the differentially expressed genes were divided into 20 functional categories, and “carbohydrate metabolism” was the most representative pathway, in which starch and sucrose metabolism were very important for pollen development [19]. Silencing of SlMYB33 disrupted pollen maturation by inhibiting the transcription of starch and sucrose metabolism-related genes and sugar transport [20]. Hyodo et al. compared transcripts of downward sepal mutant and wild type by RNA-Seq sequencing and found that a gene related to cell expansion, *XTH*, was significantly different. The *XTH* gene itself could not cause cell wall relaxation or creep but could act synergistically to promote cell wall extension, which can be called indirect or secondary loosening agents [21]. It has also been found to influence sepal morphology by controlling genes and phytohormone regulation related to cell wall biosynthesis and modification, especially auxin, gibberellin, and cytokinin [22]. Kai et al. used RNA-Seq analysis to study the function of *the SlNCED1* gene in tomato gynoecium development and fruit setting and found that the expression of most genes related to carbohydrate and lipid metabolism was significantly different during gynoecium development, suggesting that carbohydrates and lipids are essential for gynoecium development [23]. In this study, transcriptome sequencing was performed on different stages of flower development in wild-type tomatoes, and a total of 8536 differentially expressed genes were found. Through GO analysis, it was found that in the biological process categories classified by GO, cellular process, metabolic process and biological regulation were significantly enriched. KEGG pathway analysis showed that KEGG-enriched pathways mainly included plant hormone signal transduction, starch and sucrose metabolism, pentose and glucuronic acid conversion, etc. Among them, the GO classification of the bZIP family was mainly “molecular function”, and KEGG pathways mainly included plant hormone signal transduction pathway and MAPK signaling path-plant. This result is consistent with the results of previous studies, indicating that cells undergo active energy metabolism during flower development, which may be related to the formation and flower development. The biosynthesis of some substances indicates that a large number of substances are needed to provide energy for flower development. Some genes were also enriched in the plant hormone signal transduction pathway, indicating that plant hormone also have some influence on flower development.

The AGAMOUS (AG) evolutionary branch of the MADS-box gene family plays an important role in the development of stamens and carps and the regulation of later fruit ripening in tomatoes. This evolutionary branch contains two lineages, euAG and PLE. Tomato *TAG1* of the euAG lineage belongs to the class C gene of flower development, which controls placenta and seed formation, regulates stamen and carpe characteristics, and negatively regulates leus. Its expression is upregulated in the early stage of flower development and down-regulated in the late stage, and its transcript is restricted to the same floral meristem region [24,25]. *TAGL1* of the PLE lineage is a homologous gene of *AtSHP*, which is involved in cuticle development and lignin biosynthesis inhibition promotes carotene and ethylene synthesis and pericarp thickening [26]. The down-regulation of *TAG1* and *TAGL6* MADS-box genes and the activation of genes involved in photosynthesis and sucrose metabolism are indispensable regulatory components in the process of fruit setting at the later stage of flower development when the transition to the fruit setting program occurs [27]. The *TDR4* gene, an important TF for tomato fruit ripening, showed changes in the expression of genes involved in multiple metabolic pathways, including amino acid and flavonoid biosynthesis pathways. Metabolomics analysis revealed levels of various amino acids such as phenylalanine, tyrosine and organic acids, suggesting that *TDR4* is involved in tomato fruit ripening and nutrient synthesis [28]. Vrebalov, J. et al. found that *LeMADS-MC* affected sepal development and inflorescence certainty [25]. The expression levels of *TAG1* (*Solyc02g071730*) and *LeMADS-MC* (*Solyc05g056620*) selected in this study were down-regulated at the full flowering stage, which was consistent with the down-regulation of *TAG1* expression at the start of fruit setting program in previous studies. The down-regulation of *LeMADS-MC* expression is speculated to be due to the decreased expression of *LeMADS-MC* in the late stage of flower development since the sepals have been fully developed and the inflorescence has been established. In addition, the expression of another gene, *Solyc02g087860*, also decreased to a certain extent. BLAST from NCBI found that this gene was related to the formation of stigma. It was speculated that its expression pattern was similar to that of *the MC* gene, and its expression increased during the formation of stigma in the early stage of flower development and decreased after the completion of stigma development.

ABF/AREB is A member of Group A of the bZIP TFs family, including *ABF1*, *ABF2* (*AREB1*), *ABF3*, *and ABF4* (*AREB2*), which functions downstream of *SnRK2.2*, *SnRK2.6* and *SnRK2.3*. These transcription factors may be activated primarily through phosphorylation [29]. *AREB1*, *AREB2* and *ABF3* are major factors regulating drought stress responses [30]. In addition to being an important signal in drought resistance, the ABA signal also plays a regulatory role in plant growth and development. Some studies have shown that ABA can significantly accelerate the proliferation of inflorescence meristems [31]. *SlAREB1*, a close neighbor of *AREB/ABFs* in Arabidopsis, plays a role in abscisic acid signal transduction and regulates primary metabolism in tomato fruit [32]. *ABF4*, together with *ABF3* and *NY-FCs*, can induce *SOC1* to promote early flowering under drought conditions [33]. The expression level of *ABF4* selected in this study did not increase significantly at full flowering. *AtbZIP61* has been reported to be expressed in pollen [34], and *AtbZIP43* has been shown to form a heterodimer with it, suggesting a possible functional link between them [35]. In this study, three genes, *bZIP61*, *bZIP43* and *bZIP61-like*, were selected, and their expression levels were significantly increased at the full flowering stage. Among them, the expression patterns of *bZIP61* and *bZIP43* were similar, suggesting that their functions in tomato flower development were similar to those in Arabidopsis. The relative expression level of *bZIP61-like* increased significantly, and the expression level of *bZIP61-like* in the full flowering stage was twice that in the bud stage, which was different from the expression pattern of *bZIP43* and *bZIP61*, suggesting that *bZIP61* and *bZIP61-like* may have functional redundancy, and the relationship between these three TFs in tomato needs to be further characterized.

## 4. Materials and Methods

### 4.1. Sample Preparation 

The plant material AC (The wild-type tomato *Alisa Craig*, is characterized by early ripening, round red fruit with medium size and unique flavor, which is suitable for greenhouse cultivation. Height: 200 cm (79″). Spread: 50 cm) used in this study was obtained from seeds stored in our laboratory. Plant materials were planted from April to October 2021 in the solar greenhouse of the Horticulture Experimental Station of Northeast Agricultural University (126°54′58″ E, 45°46′23″ N). The age of the seed is one year. Using active organic matrix for seedling (Shang Tao 4.0, composition: charcoal soil, coconut bran, perlite, vermiculite; N + P2O5 + K2O content ≤ 4.0%, organic matter content ≥ 25%, effective viable bacteria (cfu) ≥ 0.1 million/g). The cavity tray was filled with matrix soil until the surface was smooth and then soaked with water. Small holes of 1 cm depth were dug in each hole, and one seed was sown in each hole. Then, the hole was filled with dry soil and placed in the greenhouse, and the non-woven fabric was used to moisten it. After germination, until the soil is completely dry, irrigate the first water; Every 2 to 3 days after that, the soil is dried and watered. The seedlings were divided at the age of 4 weeks and transplanted after growing to four leaves and one heart. The experiment was conducted in a completely randomized block design, with 12 plants planted in each plot and repeated three times. Water every 7 to 10 days until flowering and fruiting. No topdressing was applied throughout.

### 4.2. Sample Preparation

Flowers of wild-type tomato AC were collected at the bud stage (about 1 cm), and the full flowering stage (the petals were fully expanded) with three replicates each, and a total of six samples were collected. As Figure 1, Sampling at the bud stage was denoted CK1 (three replicates were CK1_1, CK1_2, CK1_3), and sampling at full bloom was denoted CK2 (three replicates were CK2_1, CK2_2, CK2_3). Six samples were subjected to RNA extraction and extracted with an RNA extraction kit (Takara, Beijing, China). Six copies of extracted RNA were used for library construction.

### 4.3. Construction of a Transcriptome Library

The construction process of the transcriptome library includes (1) mRNA purification: First, total RNA was processed and purified to obtain mRNA. (2) RNA fragmentation: The obtained RNA was fragmented and reverse transcribed with primers, followed by the synthesis of double-stranded cDNA. (3) The resulting duplex ends are complemented and flattened. (4) The ligation products were amplified by PCR using specific primers. (5) The PCR products were thermally transformed and cyclized to obtain a single-stranded circular DNA library. Finally, sequencing was performed.

### 4.4. RNA-Seq Data Analysis

After sequencing, the original data were obtained, and the original data were filtered to obtain the filtered data. Data filtering was performed using the filtering software SOAPnuke for statistics and Trimmomatic for filtering. Then HISAT software was used for comparison according to Kim et al.’s method [36]. Gene sequence alignment was performed according to the method of Langmead et al., and gene expression levels of samples were calculated according to the method of Li et al. [37,38]. According to the difference in gene expression levels between CK1 and CK2 groups, we used the DEGseq method to analyze differentially expressed genes [39], and the determination criteria of differentially expressed genes (DEG) were Q-value ≤ 0.001 and log2 fold change ≥ 2. The differentially expressed genes between CK1 and CK2 groups were compared to KEGG pathway database and GO database for cluster analysis and enrichment analysis. MAPMAN software was used to analyze the transcriptome data to visualize the enrichment analysis results. Cluster analysis of transcriptome data was performed using the tools of the BGI Interactive Reporting System (https://report.bgi.com, accessed on 8 October 2022).

### 4.5. qRT-PCR Verification of Related Genes

In this study, eight genes related to tomato flowering were selected for real-time fluorescence quantitative (Qrt-Pcr) verification. The NCBI website was used for specific primer design, and sleF-1α was used as an internal reference primer for Qrt-Pcr analysis. The total RNA of the samples was extracted using an RNA extraction kit (Takara, Beijing, China), and the total RNA was reverse transcribed into cDNA according to the instructions of the reverse transcription kit (Vazyme, Nanjing, China). Three technical replicates were performed for each biological replicate of each sample. Appendix A shows the primer sequence of the validation gene, and Appendix A shows the Qrt-Pcr reaction system. The assay was performed using AceQ^®^ qPCR SYBR^®^ Green Master Mix (Vazyme, Nanjing, China) as well as a qTOWER3G detection system (Analytik Jena, Germany). Gene expression was analyzed using the 2^−∆∆CT^ method [40].The normalization treatment was calculated by ∆Ct = Ct (sample) − Ct (sleF-1α); ∆∆CT = ∆Ct (sample1) − ∆Ct (sample2); relative quantification = 2^−∆∆CT^. SPSS 7.0 software was used to analyze the variance of qRT-PCR data, and the Waller–Duncan (W) method was used for comparison at the *p* < 0.05 level.

### 4.6. Screening of Candidate Genes

In this study, the transcriptome sequencing data during tomato flower development were analyzed and compared, and the candidate genes of the bZIP family that may be involved in tomato flower development were screened by comparing the gene quantification of tomato bZIP family genes during tomato flower development.

## 5. Conclusions

In this study, flower tissues were collected from wild-type tomato AC++ at the flowering stage (about 1 cm) and full flowering stage (the petals were fully expanded) for transcriptome analysis, and 8536 differentially expressed genes were screened out, including 3873 upregulated differentially expressed genes and 4663 down-regulated differentially expressed genes. Most of the differentially expressed genes in the two different stages were mainly involved in plant hormone signal transduction, starch and sucrose metabolism, pentose glucuronic acid conversion, etc., indicating that cells undergo active energy metabolism during the development of the flower, which may be related to the formation and flower development. Some genes were enriched in phytohormone signal transduction pathways, indicating that phytohormones also have some influence on flower development. Combined with previous studies, we found that the expression levels of *Solyc02g087860* and three of *bZIPs* were significantly different between the bud stage and the full flowering stage of flower development. The identified bZIP family genes related to tomato flower development in this study can complement the previous studies on the association between the bZIP family and tomato flower size. Our next work will conduct a detailed functional analysis of the identified bZIP family genes to characterize their roles in tomato flower development. In this study, we comprehensively investigated tomato flower development through transcriptome analysis, which provided new ideas for the regulation of tomato and other flower development.

## Figures and Tables

**Figure 1 ijms-23-15624-f001:**
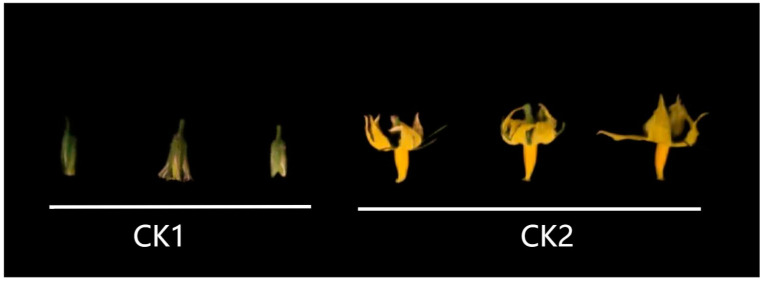
Flowers of AC at two stages. CK1: the bud stage; CK2: the full flowering stage.

**Figure 2 ijms-23-15624-f002:**
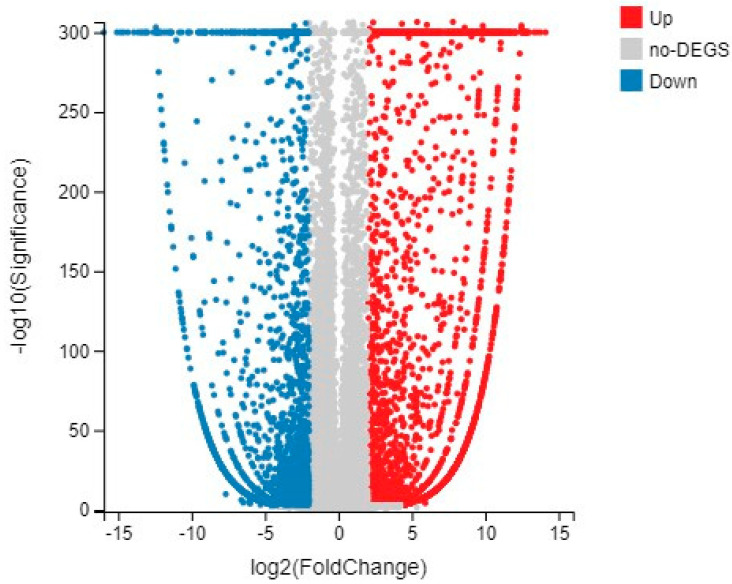
Volcanic map of differentially expressed genes between CK1 and CK2. Note: X axis: difference multiple values, Y axis: significance value.

**Figure 3 ijms-23-15624-f003:**
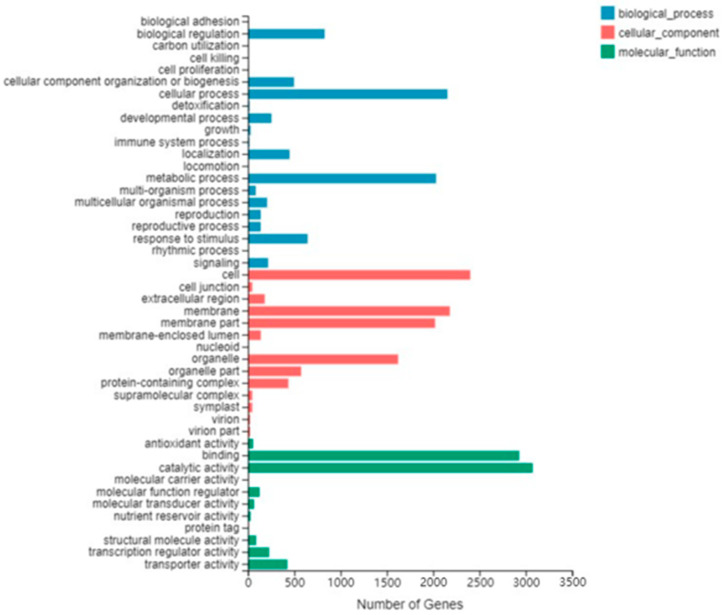
GO analysis of differentially expressed genes.

**Figure 4 ijms-23-15624-f004:**
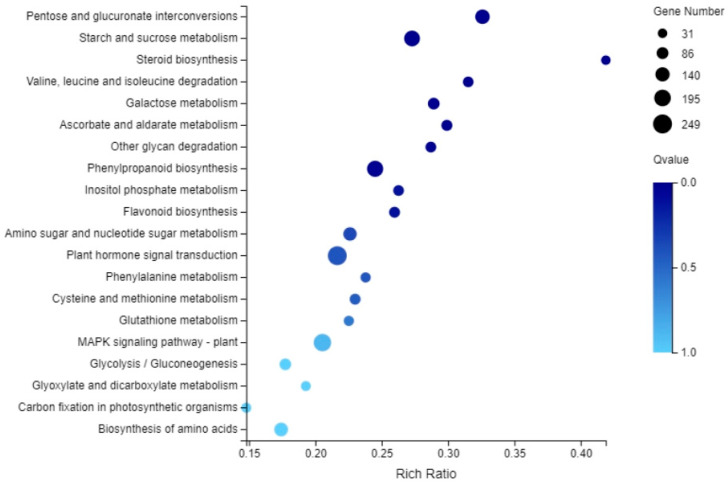
Bubble diagram of KEGG pathway enrichment of differentially expressed genes. Note: Rich Radio: enrichment ratio, the larger the bubble, the greater the number of genes.

**Figure 5 ijms-23-15624-f005:**
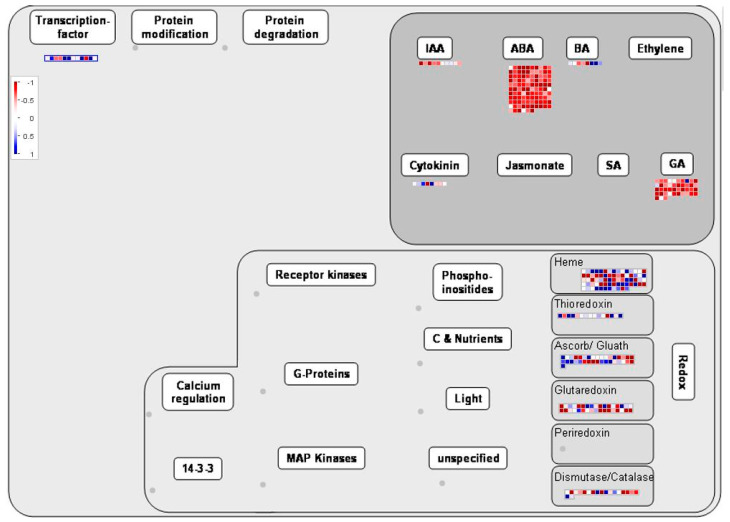
Regulation overview produced by MapMan software. Each color block represents a separate gene; red indicates upregulated expression, and the darker the color, the higher the expression. Blue indicates the down-regulation of expression, and the darker the color, the lower the expression.

**Figure 6 ijms-23-15624-f006:**
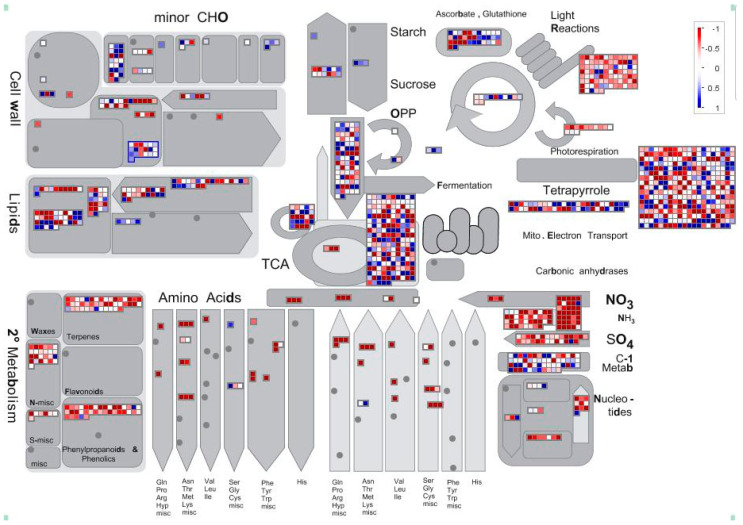
Metabolism overview produced by MapMan software. Each color block represents a separate gene; red indicates upregulated expression, and the darker the color, the higher the expression. Blue indicates the down-regulation of expression, and the darker the color, the lower the expression.

**Figure 7 ijms-23-15624-f007:**
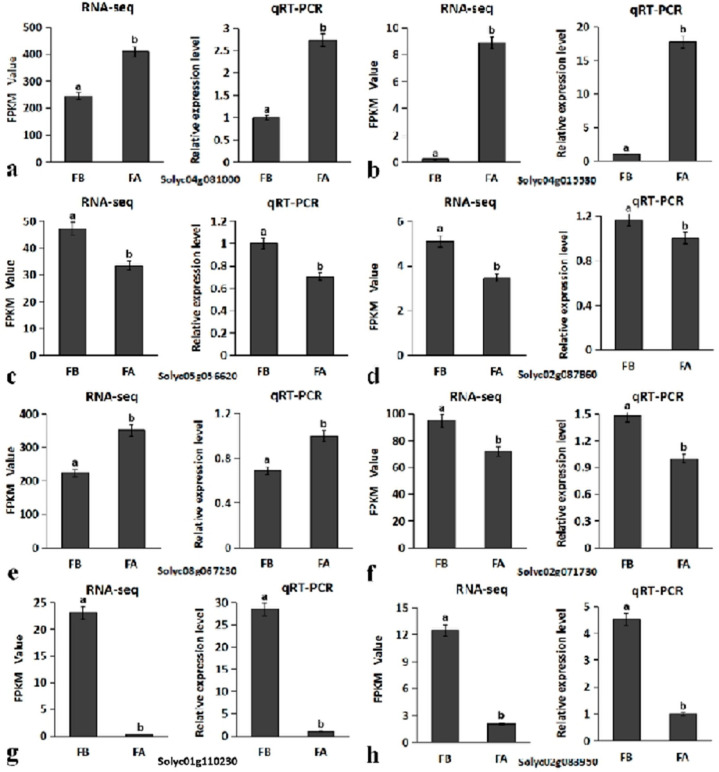
Comparison of RNA-seq and qRT-PCR expression results. FPKM values of eight DEGs were obtained by RNA-seq, and relative mRNA levels were obtained by qRT-PCR. Three technical replicates were performed for each biological replicate of each sample. Error bars indicate the standard deviation. *p* < 0.05 indicates a significant difference: a is significantly higher than b, and a significant difference between the same letters. Note: FB: flower bud; FA: full-bloom stage. Different letters indicate significant differences at the *p* < 0.05 level. Subfigures (**a**–**h**) correspond to qRT-PCR validation of the respective RNA-Seq data of the eight genes selected.

**Figure 8 ijms-23-15624-f008:**
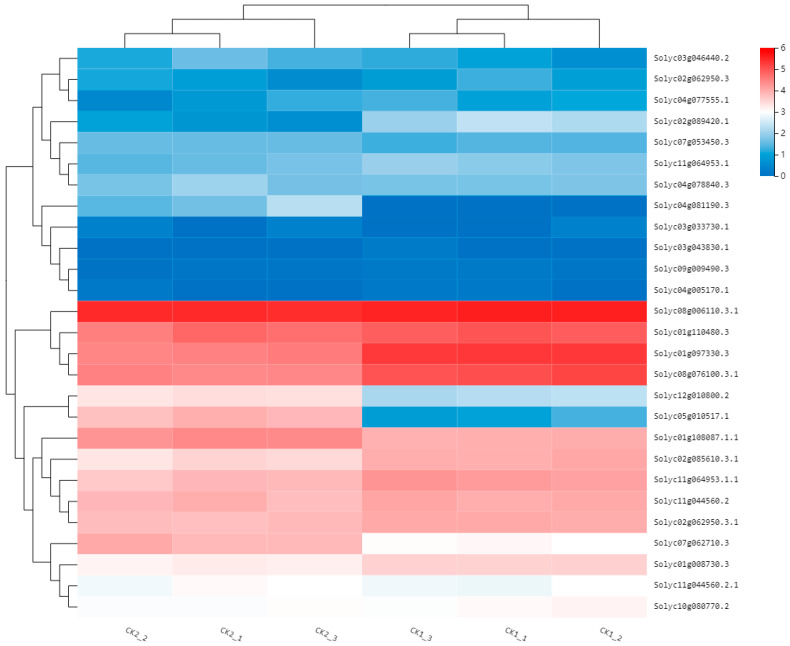
Cluster analysis of bZIP family genes expression. The horizontal axis represents the log2 (FPKM + 1) of the sample, and the vertical axis represents the gene. Under the default color matching, the more red the color of the color block, the higher the expression, and the more blue the color, the lower the expression.

**Figure 9 ijms-23-15624-f009:**
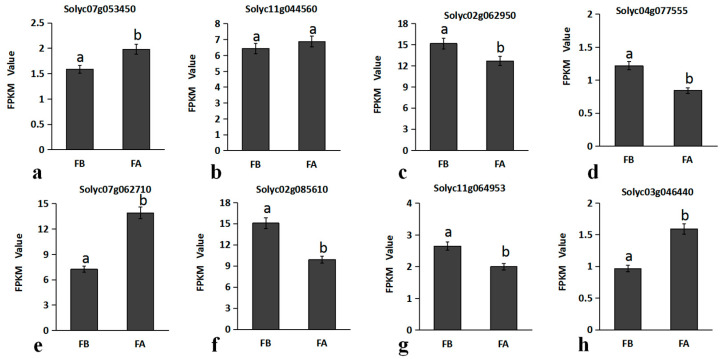
Comparative analysis of quantitative results of tomato bZIP family genes.FPKM values of eight DEGs were obtained by RNA-seq. Error bars indicate the standard deviation. *p* < 0.05 indicates a significant difference: a is significantly higher than b, and a significant difference between the same letters. Note: FB: flower bud; FA: full-bloom stage. Different letters indicate significant differences at the *p* < 0.05 level. Subgraphs (**a**–**h**) correspond to the relative expression comparison of the selected eight bZIP genes in different periods.

**Table 1 ijms-23-15624-t001:** Data quality of RNA-Seq.

Sample	Total Raw Reads (M)	Total Clean Reads (M)	Total Clean Bases (Gb)	Q20 (%)	Q30 (%)	Clean Reads Ratio (%)
CK1_1	43.82	42.94	6.44	96.12	90.89	97.99
CK1_2	43.82	43.08	6.46	95.91	90.42	98.3
CK1_3	43.82	42.95	6.44	95.97	90.55	98.02
CK2_1	43.82	42.54	6.38	96.1	90.86	97.08
CK2_2	43.82	42.9	6.44	96	90.64	97.9
CK2_3	43.82	42.78	6.42	96.05	90.73	97.61

## Data Availability

The raw sequencing data of this article are stored in the NCBI Sequence Read Archive under accession number GSE164382.

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
