# Peer review of "Transcriptome Analysis and Screening of Genes Associated with Flower Size in Tomato (Solanum lycopersicum)"

_ijms, 2022, doi:10.3390/ijms232415624_

Round 1
Reviewer 1 Report
I recommend the manuscript “Transcriptome analysis and screening of genes associated with 2 flower size in tomato (Solanum Lycopersicon)” for publication after minor revision. This research is focused to understand the molecular mechanism of flowering in tomatoes and the data of this paper is relevant with limited originality, but it could be helpful for future agricultural practices and the selection of cultivars.
Comments:
The abstract should be rewritten with more about the scientific finding of this research work.
The introduction is well written with enough evidence of past research.
The author used wild-type tomato AC++ for the research work. I will suggest giving more details about this germplasm. Please include the botanical name also.
Please avoid writing any result or conclusion in the material and method section, like in line no. 352: “It has been found that bZIP family genes are involved in the regulation of flower 352 development in Arabidopsis.”
Please include the details section for statistical analyses in material methods.
Please include the unique finding in the conclusion section.
References are okay.
Figures are okay except for 7 and 9. Please apply the statistics in figure 7 and please include which statistical test is in detail for Figures 7 and 9.
The discussion is well written.
Author Response
Response to Reviewer 1 Comments
Point 1: The abstract should be rewritten with more about the scientific finding of this research work.
Response 1: Thank you very much for your kindly advice.We have supplemented the original abstract with some findings relevant to this study. All changes in the manuscript were marked by blue color. If there is any shortage, please inform us! We will try our best to revise this manuscript.
Point 2: The introduction is well written with enough evidence of past research.
Response 2:Thank you very much for your attention.
Point 3: The author used wild-type tomato AC++ for the research work. I will suggest giving more details about this germplasm. Please include the botanical name also.
Response 3: Thank you very much for your kindly advice. Alisa Craig, is characterized by early ripening, round red fruit with medium size and unique flavor, which is suitable for greenhouse cultivation.Height: 200cm (79"). Spread: 50cm. At your suggestion, a description of plant material characteristics has been added to the manuscript. All changes in the manuscript were marked by blue color. If there is any shortage, please inform us! We will try our best to revise this manuscript.
Point 4: Please avoid writing any result or conclusion in the material and method section, like in line no. 352: “It has been found that bZIP family genes are involved in the regulation of flower 352 development in Arabidopsis.”
Response 4: Thank you very much for your kindly advice. The line no.392:"It has been found that bZIP family genes are involved in the regulation of flower development in Arabidopsis."has been removed according to your suggestion.
Point 5: Please include the details section for statistical analyses in material methods.
Response 5: Thank you very much for your kindly advice.According to your suggestions, we supplemented the details of qRT-PCR statistical analysis, including total RNA extraction; Reverse transcription; Technical and biological replicates; Methods and details of data processing.All changes in the manuscript were marked by blue color. If there is any shortage, please inform us! We will try our best to revise this manuscript.
Point 6: Please include the unique finding in the conclusion section.
Response 6: Here,some bZIP family genes were identified in tomato in our study, and they were associated with tomato flower size. These genes were previously reported in Arabidopsis but not in tomato. Our future work will conduct detailed functional validation of these genes to characterize their relationship with tomato flower size and their roles in tomato flower development. These changes have been added to the conclusions. All changes in the manuscript were marked by blue color. If there is any shortage, please inform us! We will try our best to revise this manuscript.
Thank you very much for your attention and kindly advice.
Point 7: References are okay.
Response 7: Thank you very much for your attention.
Point 8: Figures are okay except for 7 and 9. Please apply the statistics in figure 7 and please include which statistical test is in detail for Figures 7 and 9.
Response 8: Here,the application of Figure 7 is to verify the accuracy of the RNA-seq results by comparing the qRT-PCR results of the 8 randomly selected genes with the RNA-seq data. The results showed that the RNA-seq data were reliable.
Supplementary instructions for the statistical analysis have been provided for Figures 7 and 9 according to your suggestions.All changes in the manuscript were marked by blue color. If there is any shortage, please inform us! We will try our best to revise this manuscript.
Thank you very much for your attention and kindly advice.
Point 9: The discussion is well written.
Response 9: Thank you very much for your attention.
Thank you very much for your attention and kind advice.

Reviewer 2 Report
Manuscript Number: ijms-2051752, titled:
Transcriptome analysis and screening of genes associated with flower size in tomato(Solanum lycopersicum).
Review 1 – 13 November 2022
Dear Editor of International Journal of Molecular Sciences
the argument is interesting but it has to be improved. The M&M section has to be improved. The references section is not arranged as per IJMS instructions for authors. Many inaccuracies in the text;
I suggest a major revision
To the Authors (in detail):
1) the argument is interesting but it has to be improved. The M&M section has to be improved. The references section is not arranged as per IJMS instructions for authors. Many inaccuracies in the text;
2) In the title, insert one space after … tomato;
3) Verify the spacing in the lines of the corresponding author;
4) Line 32, verify the indent;
5) Line 33, sepals, in small letter, be consistent with the following names;
6) Line 34, insert one space after tomato;
7) Line 41, insert one space before… When;
8) Line 57, insert one space before class E;
9) Lines 53 and 59, the reference Koen et al is missed in the references section, in addition, the reference 9 is another one. Please, re-arrange the whole section;
10) Line 97, insert one space before… In;
11) Line 109, delete one space after the second bracket and insert one space after the dot;
12) Line 133, insert one space after .. family;
13) Line 139, verify the spacing between words and punctuation;
14) Line 152 and in the whole manuscript, sometime you have written the names of molecules in capital letter and sometime in small letter. Please, be consistent in the whole manuscript: small or capital letter?
15) Line 164, verify the spacing between words and numbers and punctuation;
16) Line 202, delete one space after the first bracket and one space before the second bracket;
17) Lines 221-225, the references 19-20 are not only referred to Zhang et al. please, re-arrange;
18) Line 307, too many commas;
19) Line 308, insert one space before … and;
20) Sub-section 4.1, please, indicate the age of seeds;
21) Sub-section 4.1, please, indicate the substrate, fertilizers (type, quantity, period), and irrigation (type, quantity, period);
22) Sub-section 4.1, describe the botanical characteristic of the plant material you have used in your experiment and the year of your experiment;
23) Line 307, verify spacing and punctuation, verify also where to write (Figure 1);
24) References section: the references are not listed as per IJMS instructions for Authors, please, re-arrange this section;
25) Please, write in blue color or evidence differently the corrections you will do.
I suggest a major revision
Regards.
Author Response
Response to Reviewer 2 Comments
Point 1: the argument is interesting but it has to be improved. The M&M section has to be improved. The references section is not arranged as per IJMS instructions for authors. Many inaccuracies in the text;
Response 1: Thank you very much for your kind advice.Changes have been made to the Materials and Methods and References sections based on your comments
Point 2: In the title, insert one space after … tomato;
Response 2: This section was modified according to your kind suggestion.
Point 3: Verify the spacing in the lines of the corresponding author;
Response 3: This section was modified according to your kind suggestion.
Point 4: Line 32, verify the indent;
Response 4: We have revised the indent.
Point 5: Line 33, sepals, in small letter, be consistent with the following names;
Response 5: Here,“Sepals” are at the beginning of a sentence, so the first letter is capitalized.
Point 6: Line 34, insert one space after tomato;
Response 6: This section was modified according to your kind suggestion.
Point 7: Line 41, insert one space before… When;
Response 7: This section was modified according to your kind suggestion.
Point 8: Line 57, insert one space before class E;
Response 8: This section was modified according to your kind suggestion.
Point 9: Lines 53 and 59, the reference Koen et al is missed in the references section, in addition, the reference 9 is another one. Please, re-arrange the whole section;
Response 9: Thank you very much for your responsible comments. Here's my mistake. The full name of the author of this reference is Koen Geuten. "Koen" in the text has been changed to "Geuten" to be consistent with the reference 9.
Point 10: Line 97, insert one space before… In;
Response 10: This section was modified according to your kind suggestion.
Point 11: Line 109, delete one space after the second bracket and insert one space after the dot;
Response 11: This section was modified according to your kind suggestion.
Point 12: Line 133, insert one space after .. family;
Response 12: This section was modified according to your kind suggestion.
Point 13: Line 139, verify the spacing between words and punctuation;
Response 13: We have revised the spacing between words and punctuation.
Point 14: Line 152 and in the whole manuscript, sometime you have written the names of molecules in capital letter and sometime in small letter. Please, be consistent in the whole manuscript: small or capital letter?
Response 14: Thank you very much for your responsible comments.Line 152 and in the whole manuscript, some revisions have been made. The sections in parentheses, based on the pictures of GO analysis and KEGG analysis in the manuscript, decided to use small letter for proper terms related to GO analysis and capital letter for proper terms related to KEGG analysis.
Point 15: Line 164, verify the spacing between words and numbers and punctuation;
Response 15: We have revised the spacing between words and numbers and punctuation
Point 16: Line 202, delete one space after the first bracket and one space before the second bracket;
Response 16: This section was modified according to your kind suggestion.
Point 17: Lines 221-225, the references 19-20 are not only referred to Zhang et al. please, re-arrange;
Response 17: Thank you very much for your kindly advice.This part has been rearranged.Reference 20 was modified and cited separately
Point 18: Line 307, too many commas;
Response 18: We have deleted some commas
Point 19: Line 308, insert one space before … and;
Response 19: This section was modified according to your kind suggestion.
Point 20: Sub-section 4.1, please, indicate the age of seeds;
Response 20: Thank you very much for your kindly advice. I have indicated the age of seeds is one year.
Point 21: Sub-section 4.1, please, indicate the substrate, fertilizers (type, quantity, period), and irrigation (type, quantity, period);
Response 21: Thank you very much for your kindly advice. I have indicated the substrate, fertilizers (type, quantity, period), and irrigation (type, quantity, period) in my manuscript.
Point 22: Sub-section 4.1, describe the botanical characteristic of the plant material you have used in your experiment and the year of your experiment;
Response 22: Thank you very much for your kindly advice. At your suggestion, a description of experimental years and plant material characteristics has been added to the manuscript.
Point 23: Line 307, verify spacing and punctuation, verify also where to write (Figure 1);
Response 23: We have revised spacing and punctuation and where to write Figure 1.
Point 24: References section: the references are not listed as per IJMS instructions for Authors, please, re-arrange this section;
Response 24: Thank you very much for your kindly advice. At your suggestion, the reference section has been rearranged in accordance with the IJMS Journal requirements for authors
Point 25: Please, write in blue color or evidence differently the corrections you will do.
Response 25: Finally, the manuscript has been revised according to comments of reviewers and editor. All changes in the manuscript were marked by blue color. If there is any shortage, please inform us! We will try our best to revise this manuscript.
Thank you very much for your attention and kind advice.

Round 2
Reviewer 2 Report
Manuscript Number: ijms-2051752, titled:
Transcriptome analysis and screening of genes associated with flower size in tomato (Solanum lycopersicum).
Review 2 – 3 December 2022
Dear Editor of International Journal of Molecular Sciences
the argument is interesting and the authors have included all my comments. I suggest the publication in the present form.
Regards.